# Detrimental Contexts in Open-Domain Question Answering

**Philhoon Oh**
KAIST AI
philhoonoh@kaist.ac.kr

**James Thorne**
KAIST AI
thorne@kaist.ac.kr

## Abstract

For knowledge intensive NLP tasks, it has been widely accepted that accessing more information is a contributing factor to improvements in the model's end-to-end performance. However, counter-intuitively, too much context can have a negative impact on the model when evaluated on common question answering (QA) datasets. In this paper, we analyze how passages can have a detrimental effect on retrieve-then-read architectures used in question answering. Our empirical evidence indicates that the current read architecture does not fully leverage the retrieved passages and significantly degrades its performance when using the whole passages compared to utilizing subsets of them. Our findings demonstrate that model accuracy can be improved by 10% on two popular QA datasets by filtering out detrimental passages. Additionally, these outcomes are attained by utilizing existing retrieval methods without further training or data. We further highlight the challenges associated with identifying the detrimental passages. First, even with the correct context, the model can make an incorrect prediction, posing a challenge in determining which passages are most influential. Second, evaluation typically considers lexical matching, which is not robust to variations of correct answers. Despite these limitations, our experimental results underscore the pivotal role of identifying and removing these detrimental passages for the context-efficient retrieve-then-read pipeline. [1]

## 1 Introduction

Knowledge-intensive NLP tasks such as open-domain question answering (Rajpurkar et al., 2016; Yang et al., 2018) and evidence-based fact verification (Thorne et al., 2018; Jiang et al., 2020) require models to use external sources of textual information to condition answer generation in response to an input. This family of tasks shares a

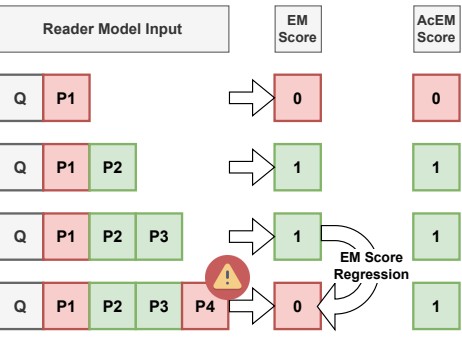

Figure 1: Even when provided with correct passage (P2) addition of further information (P4) causes the reader model to change the prediction, highlighting issues when using recall-optimized retrievers. Differences between Exact Match (EM) score and Accumulated EM (AcEM) indicate knowledge issues.

common architecture for modelling (Lewis et al., 2020; Petroni et al., 2021; Thakur et al., 2021; Izacard et al., 2022; Lyu et al., 2023) where a two-stage architecture of a *retriever* model finds contextual passages and passes these to a *reader* module that uses this context when it generates an answer.

In this two-step pipeline, expected performance (such as answer *exact match* score) increases monotonically as more information is provided to the reader model at test time. Therefore, it is commonly conceived that providing the reader with more context improves overall performance (Izacard and Grave, 2020; Liu et al., 2023). However, empirical studies (Sauchuk et al., 2022) contradict this general belief: even when true positive passages are provided to the model, the presence of relevant but non-evidential documents can change correct outputs to incorrect ones in a phenomena called *damaging retrieval*. While this phenomena was not thoroughly explored in the context of question answering, it could suggest that the current reader may be influenced by *damaging retrieval*, potentially leading to a decline in performance.

In this paper, we propose an approach to address-

---

[1] Code and data are available on https://github.com/xfactlab/emnlp2023-damaging-retrieval

ing this problem by identifying and removing contexts that are detrimental to the performance of the model. As described in Figure 1, we treat the reader model, FiD (Izacard and Grave, 2020), as a black-box oracle and identify which passages are damaging by evaluating how different subsets of the retrieved context cause the answer to change. Our experiments are conducted on two question answering tasks: Natural Questions (Kwiatkowski et al., 2019) and TriviaQA (Joshi et al., 2017), evaluating the effect of three IR systems: DPR (Karpukhin et al., 2020), SEAL (Bevilacqua et al., 2022), and Contriever (Izacard et al., 2021).

Our findings challenge the conventional assumption that more passages lead to higher performance. In fact, introducing additional relevant contexts can actually worsen the performance of the model in question answering. By excluding passages that have a detrimental effect, we observe up to 10% improvement in the exact match score under ideal conditions without requiring any architectural modifications. Notably, these gains surpass the performance improvements seen in recent works that propose FiD extensions like FiD-light(Hofstätter et al., 2022) and FiE (Kedia et al., 2022). In addition, we analyze why automatically identifying damaging contexts is challenging, even with attention-based and model-based proxies for filtering. Despite these inherent limitations, considering effectiveness and operational efficiency, our empirical findings could potentially direct the focus of open-domain NLP models and yield performance improvements when using fewer context passages.

## 2 Background

### 2.1 Knowledge Intensive NLP

To integrate external knowledge for NLP tasks such as question answering (Kwiatkowski et al., 2019; Joshi et al., 2017; Yang et al., 2018), slot filling (Levy et al., 2017), and fact verification (Thorne et al., 2018), systems employ a retrieve-then-read pipeline of two models (Chen et al., 2017). A retriever first selects the context passages over a large corpus such as Wikipedia and then feeds information to the reader to derive the answer. Consequently, improving the recall of the retriever leads to improvement of the downstream reader.

Large-scale information fusion architectures such as Fusion in Decoder (FiD) (Izacard and Grave, 2020) are able to combine information in multiple passages when decoding the answer. In contrast to the conventional T5-Model (Raffel et al., 2019), each (query, passage) pair is independently encoded, and the encoded vectors are concatenated when input into the decoder model. This allows more documents to be encoded overcoming the high memory complexity of self-attention in the encoder. Experimental results indicate that this architecture also supports recall-oriented optimization of the upstream information retrieval systems. For question-answering tasks, the answer exact match scores increased monotonically with the number of context passages used (Izacard and Grave, 2020; Liu et al., 2023).

Extensions of this architecture, such as FiD-Light (Hofstätter et al., 2022), FiDO (de Jong et al., 2022) are introduced to further increase the runtime efficiency. FiD-Light, for example, compresses passage encoder embedding into first-K vectors to overcome a performance bottleneck in the decoder. In the case of FiDO, it only keeps cross-attention on every K-th decoder layer as well as applies multi-query attention to mitigate the inference costs and memory usage. Unlike these architectures, FiE (Kedia et al., 2022), an encoder-only model, adds global encoder layers to fuse the information across multiple evidence passages. While these models significantly reduce inference time, they only result in minor increases in the answer exact match score.

Recent advances in LLMs demonstrate remarkable capabilities in natural language generation tasks (Liang et al., 2022). Within the context of question answering, Lyu et al. 2023 utilize an LLM as a reader in a retrieval-augmented generation manner (Lewis et al., 2020). However, recent studies (Zheng et al., 2023) demonstrate that answers generated from LLMs vary depending on the order of given contexts/answer choices and are sometimes lost in the middle (Liu et al., 2023). While Zheng et al. 2023 proposes a method for alleviating this order-variant issue by permutating given choices, this approach is not practical for 100 candidate contexts. Although investigating this issue falls outside the scope of our research, further research is needed to address the issue of the answer variability in LLMs, which arises from the order of provided contexts/answer choices.

### 2.2 Improving Retrieval

Two different approaches have been studied for retrieving relevant contexts: one approach employs a bi-encoder architecture, where the inner product of

query embeddings and context embeddings is used as a proxy for the relevance score. ORQA(Lee et al., 2019), DPR (Karpukhin et al., 2020) are both based on BERT-based bi-encoder architectures (Devlin et al., 2018). Contriever (Izacard et al., 2021) trains a single encoder in an unsupervised manner to represent both query and context embeddings. These encodings capture the relation between query and passages where most similar results can be identified through ranking the top-$N$ articles through inner-product search between embedded passages and the embedding of the query.

Alternatively, generatieve retrieval offers an alternative mechanism where the relationship between the query and an entity is encoded in the parametric space within a model. GENRE (Cao et al., 2020) utilizes an autoregressive sequence-to-sequence model to generate canonical entity titles for a query with constrained decoding using a prefix tree. Extending this, SEAL (Bevilacqua et al., 2022) directly predicts substrings in the corpus using an FM-index. GenRead (Yu et al., 2023) does not rely on any indexing system: LLMs are used to generate synthetic contexts. These approaches demonstrate favorable precision-recall trade-offs while requiring smaller index storage footprints.

## 2.3 Damaging Retrieval

Conventionally, retrieval systems are optimized to find maximally relevant documents to support a given user query (Ferrante et al., 2018). However, Terra and Warren (2005) demonstrates that the retriever can extract relevant but harmful documents and Sauchuk et al. (2022) also show that irrelevant or incorrect documents added to the input of the model can negatively affect its performance.

Identifying these damaging passages in a two-stage pipeline has not been extensively explored. However, one can consider reranking (Iyer et al., 2021; Kongyoung et al., 2022; Glass et al., 2022) as a process for selecting an optimal subset of retrieved passages and removing detrimental ones. However, FiD architectures scale well with many passages, enabling higher exact match scores simply by increasing the number of passages provided to the model. Given a large enough budget for passages, re-ranking this set of passages may not necessarily exclude them from the answer set. Furthermore, as further discussed in Section 3.1, the FiD model is order invariant and the ranking of passages is ignored during encoding. In architectures

such as FiD that treat retrieved evidence as a set, filtering the damaging passages is more appropriate than re-ranking the retrieval results.

## 3 Motivating Pilot Study

We demonstrate the effect of damaging passages extending Sauchuk et al. (2022) for the FiD model demonstrating that the model performance is sensitive to irrelevant context information. To select damaging context passages, we employ two methods: *random sampling* and *negative sampling*. For random sampling, we select passages from the corpus, $\mathcal{C}$, uniformly at random, while for negative sampling, we use BM25 to select passages that do not contain the correct answer but have high lexical overlaps. Results are reported in Figure 2 an Table 1. We use a context of up to 5 passages on 2539 instances from the NQ dev dataset and employ FiD-large[2] trained on NQ train set. We add between 0-4 sampled passages in addition to the gold passage from the dataset. FiD is invariant to passage ordering: we derive this analytically from lack of positional encoding between the encoded passages and validate it empirically through random permutation of the context set.

### 3.1 Effects of Simulated Damaging Passages

We compute Stability Error Ratio (Krishna et al., 2021, SER): the ratio of instances with modified outputs to additional samples, given the instance has correct predictions using one gold passage.

**Random Sampling**   Random passages should not contain information related to the query.. Where models are resilient to noise, we expect that randomly sampled additional do not drastically affect the reader model. Because of the large semantic distance between the randomly sampled texts and the gold passage, irrelevant information is not damaging. Injecting these random passages had negligable increase on answer EM (Figure 2(a)) and SER is under 3% across all the cases in Table 1. This, in fact, indicates that FiD is robust to additional random passages.

**Negative Sampling**   Passages sampled by BM25 have high lexical overlap and are semantically similar meaning to the query. We expected this to result in confusion in the model as the model uses related but insufficient information to condition answer

---

[2]https://github.com/facebookresearch/fid

| # Negative Samples | 1 | 2 | 3 | 4 |
|---|---|---|---|---|
| SER w/ random | 0.75% | 1.4% | 1.7% | 2.0% |
| SER w/ BM25 | 8.3% | 12.5% | 14.8% | 15.9% |

Table 1: Stability Error Ratio on Random/BM25 negative passages. An increase in SER implies the model's instability on negative samples.

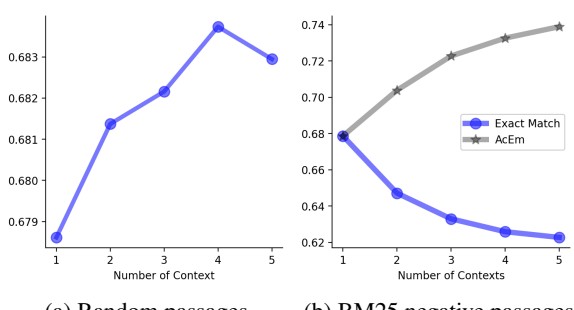

(a) Random passages     (b) BM25 negative passages

Figure 2: Exact Match scores of FiD on Natural Questions with a different number of contexts (x-axis). As the Exact Match score remains unchanged regardless of the position of the gold context, we represent this relationship using a single blue line in the plot.

generation. Adding additional negative samples results in a monotonic decrease in EM score, which is represented as a blue line in Figure 2(b) and an increase in SER, reported in Table 1. A rising number of instances where the model changes the answer from predicting the correct answer to an incorrect highlights the damaging effects.

In general, negative passages tend to have a negative impact on the model's inference. However, we have discovered that certain negative passages can actually enhance the model's performance when paired with gold evidence. To measure this, we calculate an extension of EM score called Accumulated Exact Match(AcEM). This checks the existence of Exact Match on up to top-K passages. Surprisingly, AcEM monotonically improves with additional negative passages, yielding higher results than when using only gold contexts. This demonstrates that the model can also benefit from negative samples to make correct predictions and implies that *damaging* passages should be differentiated from *negative* contexts. We report comparisons of AcEM and EM with respect to the number of context sizes in Figure 2(b).

## 4 Damaging Passages from Retrievers

Section 3 simulates the impact of damaging passages. However, these negative passages are sampled from a distribution different from what the reader is trained on. Therefore, we experiment

with retrieval models to confirm the presence of damaging effects in the retrieve-then-read pipeline.

While retrievers predict a set of passages that are relevant to the model, it does not have perfect precision, and we do not have knowledge of which passages from the retriever are beneficial or detrimental. In order to identify wheter a passage is damaging or not, we utilize the reader model as a black box oracle to distinguish damaging contexts from positive contexts. Starting with the top-1 passage, we incrementally add passages from the retrieved list and evaluate how the answer changes. If the inference is correct with the top-1, it is classified as a positive context. We then proceed to evaluate the top-1 and top-2 passages together. If those passages generate an incorrect answer, the second evidence passage will be considered damaging. This is possible because of the order invariance property of FiD. Therefore, the process can be repeated iteratively for up to the top-N passages. We then use this sequence of predictions to yield the *exact match pattern*. The exact match pattern is a binary sequence where a 1 represents that the k-th prediction matches the answer, and 0 otherwise (Figure 1). This string represents an exact match at k (EM@k) when using up to top-k passages as input. From this, we also compute accumulated exact match (AcEM). AcEM@k is the maximum of all EM@k values up to $k$. For instance, AcEM@k equals 1 if at least one of the values in (EM@1, EM@2, ..., EM@k) equals 1. Thereby, the discrepancy between AcEM@k and EM@k can be used to measure the damaging effects on retrievers.

We utilize three different retrievers for two question answering validation datasets, Natural Questions (Kwiatkowski et al., 2019, NQ) and TriviaQA (Joshi et al., 2017, TQA), which were evaluated on FiD (Izacard and Grave, 2020). Following previous work, we use a corpus of non-overlapping 100 word chunks for retrieval. We use the top-100 candidate passages retrieved by DPR[3] (Karpukhin et al., 2020), SEAL[4] (Bevilacqua et al., 2022), and Contriever (Izacard et al., 2021)[5]. For DPR, we utilize the published retrieval result, while for the other retrievers, we run the code using the default parameters. We use two published models of FiD-large that were trained on DPR retrieved contexts from NQ/TriviaQA for inference.

---

[3] https://github.com/facebookresearch/DPR
[4] https://github.com/facebookresearch/SEAL
[5] https://github.com/facebookresearch/contriever

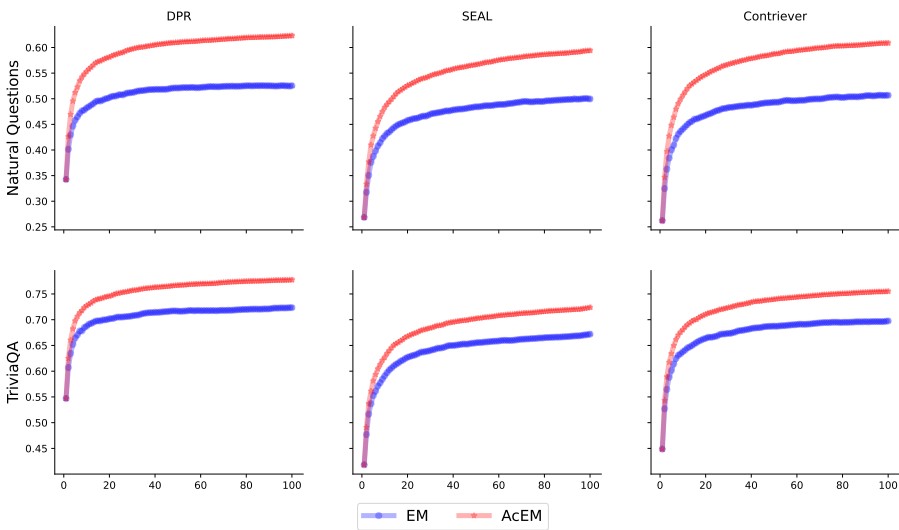

Figure 3: Incremental inference for 3 retrievers on NQ and TriviaQA dev set. This implies the existence of damaging passages in retrieved contexts and higher performance can be attained using a subset of context

Results are illustrated in Figure 3. All models show higher AcEM@k on all k values. For NQ, the EM@100 scores from DPR, SEAL, and Contriever are 52.5%, 50.0%, and 50.7% while attaining 62.3%, 59.4% and 60.8% in AcEM@100. This indicates the existence of damaging effects in retrieved contexts, and the model performance could be at least 9% points higher if the retrieved passages can be refined to remove damaging passages.

## 5 Analysis Methodology

### 5.1 Identifying Passage Types

In order to identify which passages are responsible for damaging effects in the retrieved list, we categorize passages based on the Exact Match(EM) pattern as an indicator to determine if a passage is positive or not. If the EM@k is equal to 1, it can be inferred that the kth passage is positive. However, this does not necessarily imply that the passage contains relevant information for the outcome. It could be a false positive due to the presence of preceding positive passages in the input. Similarly, a 0 in the EM pattern may suggest that the passage is negative. However, this could also be a false negative resulting from FiD's failure to predict the answer despite the presence of a correct passage.

What can be inferred from the EM pattern is when there is a change in value. If there is a transition from 0 to 1 in the EM pattern, we can infer that the passage corresponding to 1 contains at least some positive information for the outcome. Con-versely, a transition from 1 to 0 suggests that the additional passage negatively impacts the prediction. Thus, we classify passages into five types based on transitions in the EM patterns: IZ(initial zeros), DP(definite positives), DN(definite negatives), SP(semi-positive), and SN(semi-negative). Let $EM(p_k)$ denote EM@k. Then, for each $p_i$ given $\mathbf{P} = \{p_1, p_2, ..., p_N\}$, we define $\mathbf{Type}(p_i)$:

- **IZ** where $EM(p_i) = 0$ and $EM(p_j) = 0$ for all $j = 1, 2, .., i - 1$. Consecutive zeros appearing at the start of the EM pattern correspond to possibly relevant or uninformative passages.

- **DP** where $EM(p_i) = 1$ and $EM(p_{i-1}) = 0$ *or* $EM(p_i) = 1$ when $i = 1$. Definite positive passages cause EM to transition from 0 to 1 *or* the first passage enabling the correct answer.

- **DN** where $EM(p_i) = 0$ and $EM(p_{i-1}) = 1$. Definite negative passages cause a transition from a correct prediction to an incorrect one.

- **SP** and **SN** are semi-positive/negative passages do not cause transition($EM(p_i) = EM(p_{i-1})$). Incremental inference with these passages does not reveal their utility.

### 5.2 Passage Type Selection

To assess the impacts of **DN** contexts along with different context types, we generate six probe patterns to feed into the model. These patterns consist of different combinations of the **DP**, **SP**, and **SN**.

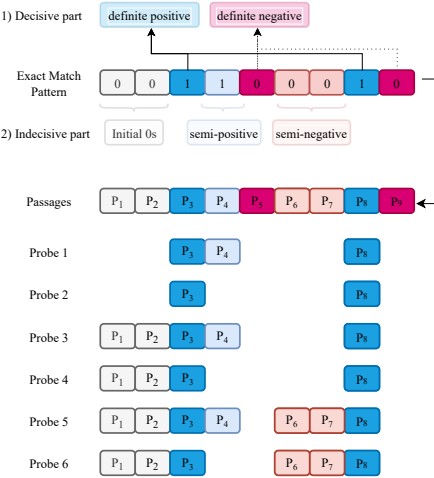

Figure 4: Context types and Probe Methods. Definite positive/negative occur on transitions of the EM pattern.

We also compare the effect of the **IZ** in our probe patterns too. In total, there are six probe patterns, illustrated in Figure 4. The most relevant approach to ours (Asai et al., 2021) only performs leave-one-out generation: a passage is considered to be positive if the model fails to generate the correct inference without the passage. This approach aids identification of positive passages but does not consider which information can be detrimental to a model. Further elaboration on the distinctions with our methodology is provided in Appendix A.2.

### 5.3 Model Analysis

In Section 5.2, we classify context types to assess their influence on the outcomes. In this section, we analyze **DP** and **DN** using two existing methods to check for distinct signals or patterns.

**Attention Score Analysis** Attention scores have been extensively studied to enhance the performance of both readers and retrievers. For instance, Xu et al. 2021 employ cross-attention mechanisms to improve the performance of extractive readers. While Izacard and Grave 2022 use attention scores from the reader as a measure of passage contribution. Building on this insight, we compare the attention scores of **DP** and **DN** passages from the reader model.

**Binary Classification** Re-ranking is commonly used for enhancing the performance of the reader in the retrieve-then-read pipeline (Iyer et al., 2021; Kongyoung et al., 2022). Additionally, Glass et al. 2022 demonstrates that a query-passage re-ranking (classification) can improve the final outcome. In-

spired by this idea, we train three different encoders to train binary classification on **DP** and **DN** contexts. This enables us to check whether the models are capable of distinguishing between **DP** and **DN**.

## 6 Experiments

### 6.1 Probe-based Selection Inference

We evaluate all 6 probing strategies from Section 5.2. Initially we add padding passages and fix the context size to 100 during the inference. For probe 3, showing the highest EM score, we further experiment with varying context sizes of 5, 10, 20 and 40 on each DPR-retrieved test set. The remaining experimental setup follows Section 4.

### 6.2 Attention Analysis

Following the approach by Izacard and Grave 2022, we calculate the cross-attention score by averaging across heads and layers of input tokens on the first output token. We conduct the experiment on top-20 DPR-retrieved contexts on NQ dev set. Next, we visualize the distribution of **DP**, **DN**, and other passage types using a density plot. We perform inference on the same dataset using passages with attention scores at various threshold values: 0.025, 0.05, 0.075, 0.1, and 0.2. We only consider contexts with an attention score higher than the threshold. If no context meets this criterion, we use the entire candidate passages instead.

### 6.3 Binary Classification

We train a binary classifier on the **DP** and **DN** contexts. To prepare the data, we split the DPR-retrieved NQ dev set into a 4:1 ratio, resulting in 7005 instances for training and 1752 instances for evaluation. From these instances, we extract 5205 **DP**s and 1516 **DN**s for training, while 1308 **DP**s and 396 **DN**s are used for evaluation. We make prediction using the question, title and context comparing a RoBERTa-large and T5-large encoder initialized with the huggingface repository checkpoints. We additionally compare using FiD-large encoder, which was pre-trained on DPR-retrieved NQ train datasets. For hyperparameters and settings, we use a batch size of 64, a learning rate of 5e-5, 3 epochs, and implement the fine-tuning process using the Huggingface transformer library[6].

---

[6]https://huggingface.co/docs/transformers

| Dataset | Retriever | EM@100 | AcEM@100 | Probe 3 |
|---------|-----------|--------|----------|---------|
| NQ | DPR | 52.5 | 62.3 | **61.8** |
| | SEAL | 50.0 | 59.4 | **52.9** |
| | Contriever | 50.7 | 60.8 | **53.0** |
| TQA | DPR | 72.3 | 77.7 | **77.6** |
| | SEAL | 67.1 | 72.3 | **72.5** |
| | Contriever | 69.7 | 75.5 | **72.5** |

Table 2: Exact Match scores using different retrievers on Natural Questions and TriviaQA development sets.

## 7 Results and Analysis

### 7.1 Probe-based Selection Inference

Using the probing patterns established in Section 5.2, we achieve significantly higher EM scores with fewer passages by only using the positive-leaning passages (Table 2). Removing **DN** and **SN** clearly demonstrates redudancy of these passages where EM@100 for probe 3 approaches AcEM@100. Scores for all other probing patterns are reported in Appendix A.1. Comparison between the other probe types indicates that the model is typically not confident with **IZ**, and it is helpful to retain this information rather than discard it.

**Few-sentence prediction** We apply Probe 3 varying the number of passages from 5, 10, 20, 40 and 100 to evaluate whether we can attain higher accuracy while using a much smaller budget of passages for the reader model. With 5 passages, the model attains more than a 12% increase in EM@5 for NQ and 8% over EM@5 for TriviaQA on the test set. Results are reported in Figure 5 and Figure 6.

While models can attain higher performance by adding more passages to their input, this comes at considerable computational costs, leading to increased memory usage and latency in prediction. Our results indicate that with just using 5 passages, state of the art EM scores can be exceeded with judicious filtering of the retrieved passages. It is crucial that retrieval models not only consider relevance for the end-user but also consider how irrelevant passages can cause a catastrophic interaction with the downstream reader model.

### 7.2 Attention Inference

Figure 7 visualizes the distribution of **DP**, **DN**, and other passage types, showing a more dispersed and higher attention score distribution of **DP**, which aligns with the previous finding (Izacard and Grave, 2022). Interestingly, however, focusing on high attention scores results in a monotonic decline in performance, as shown in Table 3. This implies

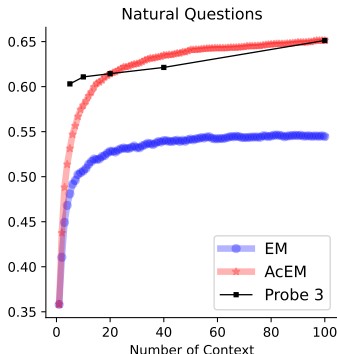

Figure 5: Probe 3 with varying size of input passages compared to incremental inference on DPR retrieved passages on Natural Questions test set.

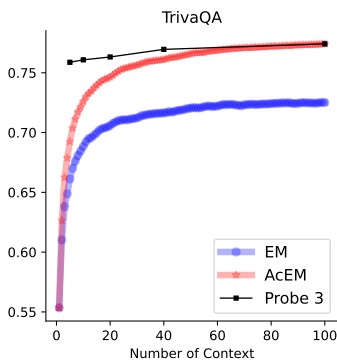

Figure 6: Probe 3 with varying size of input passages compared to incremental inference on DPR retrieved passages on TriviaQA test set.

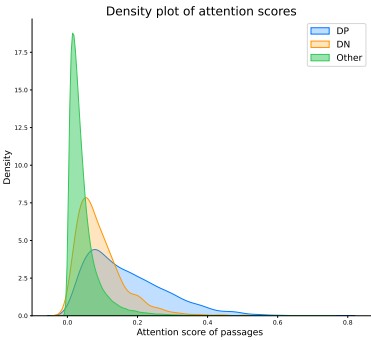

Figure 7: Attention density plot of top-20 passages in NQ development set. **DP** contexts present relatively high attention scores but cannot easily be disentangled.

that passages with high attention scores do not always guarantee the correct answer. In addition, we investigate the highest attention score in the presence of **DN**. Table 4 illustrates **DP** accounts for 40 % of the highest attention while **DN** only represents 26%. However, 67 % of transitioned answers are located within **DN**, while only 13 % are attributed to

| Attention score threshold | Exact Match |
|---|---|
| 0.025 | 49.5 |
| 0.05 | 49.11 |
| 0.075 | 48.30 |
| 0.1 | 47.55 |
| 0.2 | 46.59 |
| AcEM@20 | 58.11 |
| EM@20 | 50.21 |

Table 3: EM score of various threshold values on top-20 passages. In cases where none of the contexts met the threshold values, all top-20 passages were used.

| Feature | Context Types | Percentage |
|---|---|---|
| Highest Attention Score | DP | 40.6 |
| | DN | 26.3 |
| Transformed Prediction | DP | 13.6 |
| | DN | 67.7 |

Table 4: Out of **DN** 1091 cases, 40.6 % **DP** still exhibits the highest attentions score, but 67.7 % of transitioned predictions comes from **DN**.

**DP**, further highlighting an ambiguous relationship between attention and answers.

### 7.3 Binary Inference

We observe that it is challenging for models to differentiate between damaging contexts and positive ones. Table 5 presents the results of binary classifications. No model trained well: all models exhibit a high recall rate at the expense of precision, suggesting a strong tendency to erroneously classify most passages as **DP**s. We conjecture that the retrieved passagess already demonstrate a high relevance score over the query, so model cannot differentiate them properly.

## 8 Ablations

### 8.1 Passage Type Classification

Binary classification of passage type as **DP** or **DN** (Section 7.3) was not infomrative. We further consider passage classification as a multi-class classification on all context types and evaluate its end-to-end performance. We use the same models and datasets as in Section 6.3, resulting in 700,500 and 175,200 instances for training and evaluation, respectively. For the end-to-end evaluation, we employ the NQ test set. All training is performed with a batch size of 64, a learning rate of 5e-5, 11,000 steps (approximately 1 epoch). Multi-class classification results indicate that all models struggle with distinguishing between **DP** and **DN**, favoring predictions of **SP**, which constitutes 50% of the instances. Detailed results are reported in Appendix A.4.1 which show that all probe meth-

| Model | Acc | Pre | Re | F-1 |
|---|---|---|---|---|
| RoBERTAa (Large) | 0.76 | 0.76 | 1.0 | 0.76 |
| FiD Encoder (Large) | 0.74 | 0.77 | 0.94 | 0.85 |
| T5 Encoder(Large) | 0.75 | 0.76 | 0.987 | 0.86 |

Table 5: Binary classification results of different models on the development set. All three models are highly likely to predict **DP**s.

ods exhibited either lower or equal performance compared to EM@100, falling significantly below Probe 3. Both ablations underscore the challenge in models for predicting EM patterns and passage types, reflecting the difficulty in distinguishing **DP** and **DN** within the retrieved contexts.

### 8.2 Qualitative Analysis on Definite Negative

Results from Section 7.1 indicate that removing **DN** and **SN** passage types can significantly enhance the EM score. However, this approach relies on the availability of known answers to identify the damaging passage types. To discern patterns in the transition of EM patterns, we manually examine instances containing **DN** passages. We sample 100 examples for NQ development set and discover that out of these instances, 51 cases identified as correct answers, misled by dataset issues such as **Equivalent answers**, where answers that are semantically equivalent but have variations on surface form (e.g. alternative spellings), **Alternative answer** where only one answer out of a set is labelled in the dataset (e.g. multiple actors play a character), and **Temporality** where the correct answer depends on the current context (e.g. questions asking about latest events).

This finding supports Bulian et al. 2022 that the EM may not fully capture the impact of issues like **Equivalent answer**. The remaining 49 cases are indeed affected by **DN**. This analysis highlights the limitations of inferring damaging contexts without answers. For more details and examples of qualitative analysis on **DN**, please refer to Appendix A.3.

### 8.3 Semantically Equivalent Answers

Contemporaneous work from Kamalloo et al. 2023 asserts the limitation of a lexical matching system (EM score) that leads to the underestimation of Reader's performance. They demonstrate that prompting LLM to assess the final outcome yields similar results to manual evaluation. Inspired by this, we re-evaluate our results via zero-shot prompting, following the methodology outlined in Kamalloo et al. 2023. We calculate adjusted

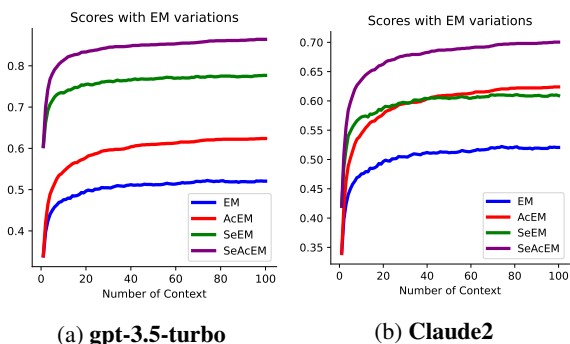

(a) **gpt-3.5-turbo**  (b) **Claude2**

Figure 8: Comparison of different metrics on NQ dev subsets, comprising 1752 instances. SeEM and AcEM are adjusted versions of EM, correcting semantically equivalent answers via LLM prompting.

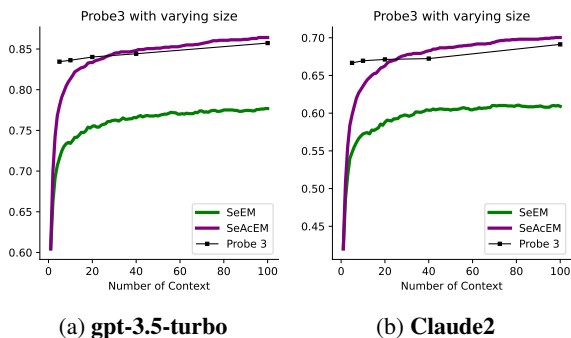

(a) **gpt-3.5-turbo**  (b) **Claude2**

Figure 9: Probe3 evaluations with varying sizes(5, 10, 20, 40, 100) on NQ dev subsets. Probe3 can achieve a performance close to the SeAcEM@100 with 20 times fewer contexts.

EM and AcEM, which refer to as SeEM(Semantic Equivalent EM) and SeAcEM(Semantic Equivalent AcEM), to assess the damaging effects. Our experiments are conducted on a subset of NQ dev set (1752 instances) using **Claude2** and **gpt-3.5-turbo** (detailed in Appendix A.5).

Figure 8 reports the results of SeEM and SeAcEM in comparison to EM and AcEM. Increase in performance were observed for both **Claude2** and **gpt-3.5-turbo** to evaluate semantic equivalence. There was a 25.6% increase in the **gpt-3.5-turbo** setting (EM@100: 0.520 to SeEM@100: 0.776) and an 8.9% increase in the **Claude2** setting (EM@100: 0.520 to SeEM@100: 0.609). However, discrepancies of 8.8% and 9.1% persist between SeAcEM@100 and SeEM@100 in both settings, indicating that damaging effects still remain despite the semantic equivalent adjustments. To evaluate the effectiveness of our Probe3, we apply the same procedure outlined in Section 7.1. The results reported in Figure 9 demonstrate a 5.8% and 5.7% increase Probe3@5 (0.834, 0.666) compared to the conventional approach SeEM@100 (0.776, 0.609). Notably, this improvement is achieved using 1/20th of the context. This result illustrates the persistent presence of damaging passages even after adjusting for semantic equivalence, emphasizing the need for filtering out damaging passages.

## 9 Conclusions

The reader models in retrieve-then-read pipelines are sensitive to the retrieved contexts when generating answers. Damaging passages in this set can lead to incorrect responses. Filtering damaging passages results in increases in EM scores without the need for architectural modifications. Despite shortcomings in evaluating QA with exact match, we demonstrate that by filtering passages, models can achieve 10% higher EM scores using subsets of context that are 20X times smaller.

## 10 Limitations

Identifying the behavior of black-box models is challenging. While we identify different subsets of evidence that the model reacts well to when generating a correct answer, there is no guarantee that these subsets of information correspond to what humans users would consider useful. Furthermore, some of the reasons the model changed its prediction, such as generating a more specific answer, would be correct if multiple references were available for evaluating the models. However, these alternative answers are not available in the datasets, which means we are optimizing the models for a limited subset of truly valid answers. Lastly, our approach may not be practical for decoder-only LLMs where the order of context/answer choices varies the outcome. To assess the answerability of the given $n$ candidate contexts, $\mathcal{O}(n!)$ inferences are required for LLMs, while FiD only needs one inference due to its order-invariance property.

We report a limitation, evaluation and position of established modelling techniques that can help guide the community for future research. If models can effectively leverage external information, they should be capable of using text as an interpretable source of information rather than relying solely on knowledge that is stored within inaccessible model parameters. This approach may contribute to a future with NLP models that are more interpretable and controllable.

# 11   Acknowledgements

This work was supported by Institute of Information & communications Technology Planning & Evaluation (IITP) grant funded by the Korea government (MSIT) (No.2019-0-00075, Artificial Intelligence Graduate School Program (KAIST)) and Artificial intelligence industrial convergence cluster development project funded by the Ministry of Science and ICT(MSIT, Korea) & Gwangju Metropolitan City.

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

## A  Appendix

### A.1  Probe pattern performance

We report the performance of all probe patterns from Section 5.1. Probe pattern 3 gives EM scores that are closest to the established AcEM@100 upper bound.

| Dataset | Retriever | EM@100 | AcEM@100 | Probe 1 | Probe 2 | Probe 3 | Probe 4 | Probe 5 | Probe 6 |
|---|---|---|---|---|---|---|---|---|---|
| NQ | DPR | 52.5 | 62.3 | 58.1 | 56.3 | **__61.8__** | 61.7 | 55.2 | 56.0 |
| | SEAL | 50.0 | 59.4 | 43.9 | 30.0 | **52.9** | 44.9 | 52.5 | 44.6 |
| | Contriever | 50.7 | 60.8 | 44.3 | 29.0 | **53.0** | 45.1 | 52.6 | 44.4 |
| TQA | DPR | 72.3 | 77.7 | 76.2 | 75.1 | **__77.6__** | 77.4 | 73.8 | 74.0 |
| | SEAL | 67.1 | 72.3 | 67.9 | 51.5 | **72.5** | 63.7 | 72.3 | 63.9 |
| | Contriever | 69.7 | 75.5 | 68.4 | 51.7 | **72.5** | 62.9 | 72.3 | 62.9 |

Table 6: Exact Match scores using different retrievers on Natural Questions and TriviaQA development sets. The datasets retrieved by DPR exhibit a consistently high EM score across various Probes. This observation can be explained by the fact that the FiD models used for evaluation are trained on DPR retrieved datasets.

### A.2  Difference between leave-one-out masking and EM pattern

There are two key distinctions between Asai et al. 2021 and our apporach. Firstly, while leave-one-out masking focuses on finding positive contexts, ours method(Passage Type Selection) aims to detect detrimental passages that can negatively impact the inference and it can also distinguish positive contexts effectively. Secondly, the time taken for inference to determine whether a passage is positive or negative is faster in our approach compared to leave-one-out masking. For instance, let $N-1$ be the number of contexts, $\mathcal{G}$ denote the reader model, and $\mathbf{P} = \{p_1, p_2, ..., p_{N-1}\}$ denote the set of retrieved passages for a given query. When new context, $p_N$, is added to the passages lists, it requires N inferences over N-1 passages in the case of leave-one-out masking, whereas ours requires a single inference over N passages.

### A.3  Qualitative Analysis

In this section, we delve further into the qualitative analysis of **DN**. We focus on the top-20 passages as examining the top-100 passages would require a more extensive investigation. Out of the 8,757 cases in the NQ development set, 1,091 instances exhibit the presence of **DN**, accounting for approximately 12% of the dataset. We selected a sample of the first 100 cases and discovered that out of these instances, 51 **DN** cases were attributed to limitations within the dataset, which can be regarded as false negative cases. Other 49 cases are true-negative cases and occurred due to the presence of **DN**. Regarding the limitations within the dataset, we classify them into three distinct types: **Equivalent Answer Example**, **Alternative Answer**, and **Temporality**.

#### A.3.1  Dataset Issue

Dataset issues exemplify situations where the predicted answers are accurate, yet the evaluation is insufficient. Prediction within the context is highlighted in red, while the supporting information is marked in blue.

- **Equivalent Answer Example**
  There are 34 cases where predictions have different formats or are supersets/subsets of gold answers.

---

**Query**: when did the king kong ride burn down
**EM pattern**: 01000000000000000000
**Gold Answers**: ['2008']

---

**DN index**: 2
**Prediction** : June 1, 2008
**Predicted answer in DN context**: Yes
**DN context**

{'id': '4215890', 'title': 'King Kong', 'text': 'and Universal Orlando Resort in Orlando, Florida. The first King Kong attraction was called King Kong Encounter and was a part of the Studio Tour at Universal Studios Hollywood. Based upon the 1976 film "King Kong", the tour took the guests in the world of 1976 New York City, where Kong was seen wreaking havoc on the city. It was opened on June 14, 1986 and was destroyed on June 1, 2008 in a major fire. Universal opened a replacement 3D King Kong ride called "" that opened on July 1, 2010, based upon Peter Jacksonś 2005 film "King Kong".'}

- **Alternative Answer Example**
  There are 13 cases in which the predictions can serve as alternative answers for the given query

  **Query**: who introduced the system of civil services in india
  **EM pattern**: 00011111111100000000
  **Gold Answers**: ['Charles Cornwallis']

  **DN index**: 12
  **Prediction** : Warren Hastings
  **Predicted answer in DN context**: No

  **Context index including prediction**: 3
  **Context**
  {'id': '14394957', 'title': 'Civil Services of India', 'text': "administer them. The civil service system in India is rank-based and does not follow the tenets of the position-based civil services. In 2015, the Government of India approved the formation of Indian Skill Development Service. Further, in 2016, the Government of India approved the formation of Indian Enterprise Development Service. Warren Hastings laid the foundation of civil service and Charles Cornwallis reformed, modernised and rationalised it. Hence, Charles Cornwallis is known as the 'Father of Civil Service in India'. He introduced Covenanted Civil Services (Higher Civil Services) and Uncovenanted Civil Services (Lower Civil Services). The present civil services of India"}

- **Temporality Example**
  There are 4 cases where queries ask for the up-to-date context but generate the answer on the outdated context

  **Query**: who plays cat in beauty and the beast
  **EM pattern**: 01111111111111111110
  **Gold Answers**: ['Kristin Kreuk']

  **DN index**: 19
  **Prediction** : Linda Carroll Hamilton
  **Predicted answer in DN context**: No

  **Context index including prediction**: 6
  **Context**
  {'id': '728625', 'title': 'Linda Hamilton', 'text': 'Linda Hamilton Linda Carroll Hamilton (born September 26, 1956) is an American actress best known for her portrayal of Sarah Connor in "The Terminator" film series and Catherine Chandler in the television series "Beauty and the Beast" (1987-1990), for which she was nominated for two Golden Globe

Awards and an Emmy Award. She also starred as Vicky in the horror film "Children of the Corn". Hamilton had a recurring role as Mary Elizabeth Bartowski on NBCś "Chuck". Hamilton was born in Salisbury, Maryland. Hamiltonś father, Carroll Stanford Hamilton, a physician, died when she was five, and her mother later married'}

---

### A.3.2 Definite Negative Cases

The presence of **DN** contexts destabilize the reader, leading to the conversion of correct gold answer. The prediction present within the context is highlighted in red.

- **Predictions in DN context**
  In 43 cases, the predictions (transitioned from gold answers) are found within the DF passage.

  ---

  **Query**: real name of gwen stacy in amazing spiderman
  **EM pattern**: 11110011111111111111
  **Gold Answers**: ['Emma Stone']

  ---

  **DN index**: 4
  **Prediction** : Mary Jane Watson
  **Predicted answer in DN context**: Yes
  **DN context**
  {'id': '1283490', 'title': 'Gwen Stacy', 'text': 'relationship with chemical weapon developer Norman Osborn. Mary Jane Watson, a popular actress in this reality, played Gwen Stacy in the film adaptation of Spider-Man's life story. Gwen and her father read textual accounts of their deaths in the main universe, though they believe this simply to be the morbid imaginings of Peter Parker, who is suffering from mental health issues. Gwen Stacy first appeared in "Marvel Adventures Spider-Man" #53 as a new student of Midtown High. She had transferred from her previous school after the Torino Gang, a powerful New York mob, began harassing her in an attempt to ' }

  ---

- **Predictions in previous context**
  There are 4 cases where the prediction appears in the previous contexts.

  ---

  **Query**: who will get relegated from the premier league 2016/17
  **EM pattern**: 00000100011111101111
  **Gold Answers**: ['Middlesbrough', 'Sunderland', 'Hull City']

  ---

  **DN index**: 15
  **Prediction** : Norwich City
  **Predicted answer in DN context**: No

  ---

  **Context index including prediction**: 7
  **Context**
  {'id': '19245453', 'title': '2016–17 Premier League', 'text': "the league – the top seventeen teams from the previous season, as well as the three teams promoted from the Championship. The promoted teams were Burnley, Middlesbrough and play-off winners Hull City, who replaced Aston Villa, Norwich City and Newcastle United. West Ham United played for the first time at the London Stadium, formerly known as the Olympic Stadium. Although having a capacity of 60,010, for the first Premier League game this was limited to 57,000 due to safety fears following persistent standing by fans at West Ham's

Europa League game played in early August. Stoke City announced that from"}

---

- **Predictions not in contexts**
  There are 2 cases where the prediction does not exist in the candidate passages.

---

**Query**: how many episodes does riverdale season one have
**EM pattern**: 11110000111111111111
**Gold Answers**: ['13']

---

**DN index**: 4
**Prediction** : 21
**Predicted answer in DN context**: No

---

**Context index including prediction**: None

---

## A.4 Passage Type Classification

### A.4.1 Mutli-classification

| Model | Metric | DP | DN | SP | SN | IZ |
|---|---|---|---|---|---|---|
| FiD-Encoder | precision | 0 | 0 | 0.54 | 0.09 | 0.5 |
| | recall | 0 | 0 | 0.72 | 0.04 | 0.37 |
| T5-Encoder | precision | 0 | 0 | 0.55 | 0.1 | 0.5 |
| | recall | 0 | 0 | 0.72 | 0.05 | 0.37 |
| RoBERTa | precision | 0 | 0 | 0.5 | 0 | 0 |
| | recall | 0 | 0 | 1 | 0 | 0 |
| Total Instances | | 1308 | 396 | 86936 | 15438 | 71122 |

Table 7: The table shows the results on 175,200 evaluations instances, which are 20% of original DPR-retrieved NQ devset. All models struggle with identifying **DN** and **DP**, favoring predictions of **SP**, which constitutes 50% of instances. Notably, all three models can't differentiate context types properly. Among 175,200 total instances, T5 predicts 114,606 (65.4%) to SP, FiD labels 115,522 (65.9%) as SP, and RoBERTa classifies all as SP. This highlights multi-classification's challenge in capturing passage types and inability to identify **DN** and **DP**.

### A.4.2 End-to-End Results

| Model | Probe 1 | Probe 2 | Probe 3 | Probe 4 | Probe 5 | Probe 6 |
|---|---|---|---|---|---|---|
| FiD-Encoder | 48.59 | 37.51 | 53.77 | 42.74 | 54.71 | 43.66 |
| T5-Encoder | 48.73 | 37.45 | 53.46 | 42.3 | 54.24 | 43.19 |
| RoBERTa | 54.43 | 35.84 | 54.43 | 35.84 | 54.43 | 35.84 |

Table 8: End-to-End result of multi-classifcation on NQ test set. Among all classification models, RoBERTa achieves the highest EM scores across Probe1, Probe3, and Probe5. This is due to the fact that RoBERTa inferences are all **SP**s, resulting in all contexts being used as inputs for the reader. Consequently, these scores should match EM@100(54.43), which uses all contexts for inference. All Probe results exhibit either lower or equal performance compared to EM@100, notably falling significantly below Probe 3's EM score(65.12). This underscores the classification models' struggle in identifying EM patterns and passage types, reflecting the challenge in distinguishing **DN** and **DP** within the provided retrieved context.

## A.5 Semantically Equivalent Answers

To calculate the SeEM and SeAcE, candidate answers are collected from the incremental answers. Then, we prompt **gpt-3.5-turbo-0613** and **Claude2** by iterating the candidate answer list over the gold answer list. Suppose we have an instance like this:

```
query = "who sings does he love me with reba"
gold_ans_lst = ['Linda Davis']
cand_lst = ['Linda Kaye Davis', 'Linda Davis']
```

In this case, EM fails to capture "Linda Kaye Davis" as the correct answer because of "Kaye" in the middle. We perform zero-shot-prompting by iterating over **gold_ans_lst** and **cand_lst**. Here is an example:

```
Question: who sings does he love me with reba
Answer: Linda Davis
Candidate: Linda Kaye Davis
Is candidate correct?
```

Although each API generates different response formats to the given prompt, we only consider cases where the response starts with explicit tokens (Yes/No); otherwise, we discard the candidate.

```
response: Yes, the candidate is correct.
Linda Kaye Davis is the singer who performed the duet "Does He Love You" with Reba McEntire.
```

Finally, we construct an adjusted gold answer list based on the response and calculate the SeAm and SeAcEm.

## A.6 Attention Score of top-20 passges

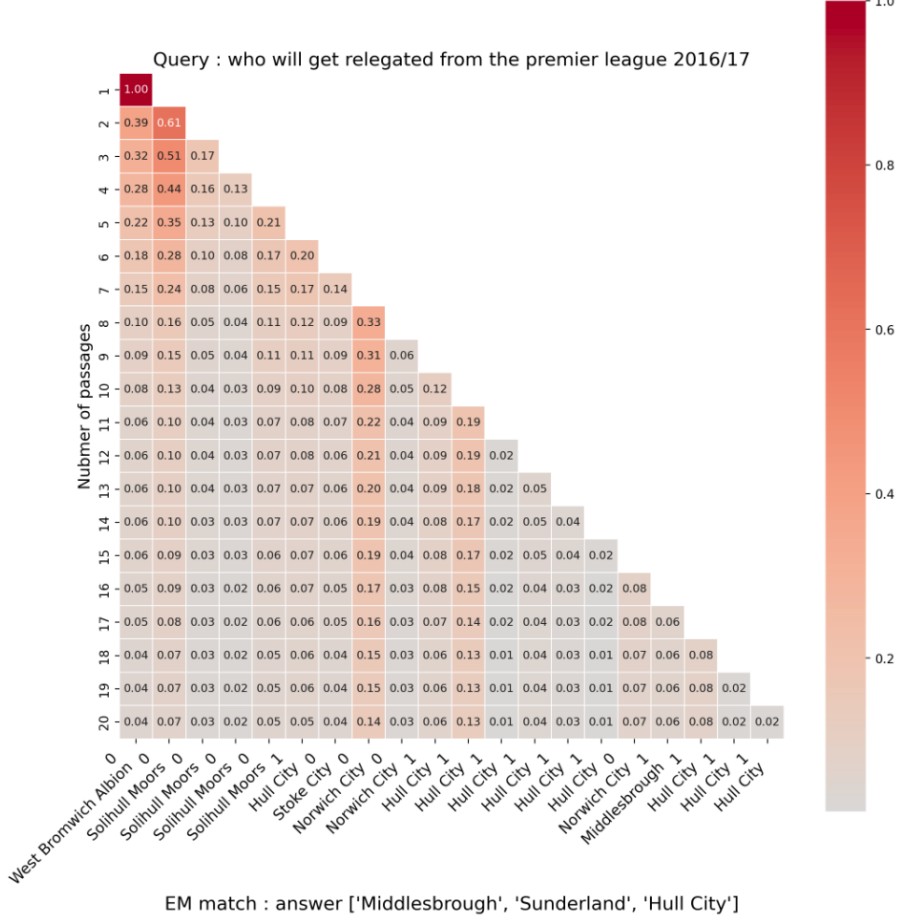

Figure 10: The attention scores of the top 20 retrieved passages are depicted in this example. It can be observed that the 7th and 16th passages are **DN**. Interestingly, the passage with the highest attention score(8th) is identified as **SN**