# OpenReview forum: "Detrimental Contexts in Open-Domain Question Answering"
_EMNLP/2023/Conference — EMNLP 2023 Findings_

### Official Review · Reviewer_LMuo · 2023-07-30

**Soundness:** 2

**Excitement:**

4: Strong: This paper deepens the understanding of some phenomenon or lowers the barriers to an existing research direction.

**Paper Topic And Main Contributions:**

This paper proposes a method to identify and filter damaging passages in the retrieve-then-read architectures, which improves at least 10% on accuracy for two popular QA tasks. The method is motivated by the fact that the model performance could be hurt by the incremental inference on BM25 negative samples. Thus, the paper defines 5 types of passages based on their EM results compared to previous retrieved passages. It is observed that a 10% performance gain can be achieved by removing definite negative and semi-negative passages.

**Questions For The Authors:**

1. Figure 4: DF -> DP in the legend
2. Line 569: This finding supports ? that
3. Why binary classification is considered instead of multi-class classification on the 5 types? Why not apply the classification results to the passage selection to see the end-to-end result?

**Reasons To Accept:**

There are many interesting insights observed from the experimental results. For example,  passages with low attention scores can contribute to the correct predictions, while passages with high attention scores do not always guarantee the correct answer. For another example, retrieval models should not optimize for the recall but also consider how irrelevant passages can cause a catastrophic interaction with the downstream reader model. These insights can be helpful for future research.


**Reasons To Reject:**

1. The writing of this paper needs improvement. First, the organization is poor. There is no introduction on why we need Selection Inference, Attention Analysis, and Binary Classification. Second, the content is not properly prioritized. A whole section (section 3 Damaging Retrieval) with more than one page is used to discuss the effect of damaging passages, which is a known phenomenon in previous works and contributes little to the introduction of the main contribution of the paper. Third, it lacks the necessary details. For example, how the binary classification works? Does it consider the question as input? Are the models initialized with the trained reader on the QA dataset?
2. The paper still does not answer the question: why more information can be detrimental to a model. Instead, the paper simply presents some unexpected observations without explaining them with a hypothesis.
3. The 10% performance gain is achieved in an ideal scenario. Identifying the definite negative and semi-negatives is not a trivial problem.

**Reproducibility:**

4: Could mostly reproduce the results, but there may be some variation because of sample variance or minor variations in their interpretation of the protocol or method.

**Reviewer Confidence:**

3: Pretty sure, but there's a chance I missed something. Although I have a good feel for this area in general, I did not carefully check the paper's details, e.g., the math, experimental design, or novelty.

---

> ### Author Rebuttal · Authors · 2023-08-29
>
> Dear Reviewer, LMuo
>
> Thank you for your appreciation of our work and insightful review. We are grateful that you find our problem, identifying damaging passages, crucial. We are also encouraged that you see the potential of our proposed insights to have an impact on future research. We hope that the final version of our paper will more clearly communicate our important findings.
>
> Below, we have tried our best to address your questions and concerns. We have addressed each "Reason to Reject" and "Question for the Authors" as denoted by R1, R2, Q1, Q2, and so forth. If any subsections or questions are present within these, they will be referenced as R1-1, R1-2, and so on. Please take a look at our response and let us know if further clarification is needed.
>
> **R1) The writing of this paper needs improvement. First, the organization is poor. There is no introduction on why we need Selection Inference, Attention Analysis, and Binary Classification. Second, the content is not properly prioritized. A whole section (section 3 Damaging Retrieval) with more than one page is used to discuss the effect of damaging passages, which is a known phenomenon in previous works and contributes little to the introduction of the main contribution of the paper. Third, it lacks the necessary details. For example, how the binary classification works? Does it consider the question as input? Are the models initialized with the trained reader on the QA dataset?**
>
> We address R1) into 3 responses.
>
> **R1-1) First, the organization is poor. There is no introduction on why we need Selection Inference, Attention Analysis, and Binary Classification.**
>
> We appreciate the feedback. We'll re-organize each section and incorporate introductions and motivations behind selection inference, attention analysis, and binary classification. We will dedicate additional space in section 4 to describe these methods and provide a clearer description of why these methods were used. Furthermore, we will clearly separate the experimental findings and further analyses and provide the end-to-end evaluation.
>
> **R1-2) Second, the content is not properly prioritized. A whole section (section 3 Damaging Retrieval) with more than one page is used to discuss the effect of damaging passages, which is a known phenomenon in previous works and contributes little to the introduction of the main contribution of the paper.**
>
> Damaging passages have not been thoroughly explored within the context of open-domain question answering. It is a fairly new concept, and we provide novel results in addition to (Sauchuk et al. 2022).
>
> The limitation of previous work is that all sentences are jointly encoded in a single model and are not representative of open-domain question answering using FiD architecture. Our findings show that the issue is beyond joint encoding of passages, which was also reported in the [Lost in the Middle: How Language Models Use Long Contexts](https://arxiv.org/abs/2307.03172)  (Liu, Nelson F., et al.,2023) but also extends to independent encoding with FiD.
>
> This allows us to affirm the invariance of FiD on the set of contexts as well as the existence of damaging passages in Reader, which allows us to perform incremental inference. This has not been addressed in previous works.
>
> To aid with the readability and prioritization of the paper’s content, we will merge The knowledge-intensive NLP and damaging passage sections as “previous work” and separate our simulating damaging passages experiments (previously in 3.1 and 3.2) as an experimental finding to make it clearer that this is a key contribution.
>
> **R1-3)  Third, it lacks the necessary details. For example, how the binary classification works? Does it consider the question as input? Are the models initialized with the trained reader on the QA dataset?**
>
> We will add necessary information in selection inference, attention analysis, and binary classification (5.1, 5.2, 5.4) in the experiment section. Regarding questions in R1-3), here are the details:
>
> - Binary classification is trained on DP and DN as described in section 5.3. We will add motivation and explanation in section 4. Method for clarification.
>
> - Input format is the same as that of FiD’s, “question : {question} title: {title} context : {passage}” for all classification methods.
>
> - RoBERTa(Large) and T5 Encoder(Large) are initialized with the huggingface repository checkpoints, and we employ FiD Encoder (Large) embeddings available on its source repository, which was pre-trained on DPR retrieved NQ train datasets.
>
>
> **R2) The paper still does not answer the question: why more information can be detrimental to a model. Instead, the paper simply presents some unexpected observations without explaining them with a hypothesis.**
>
> Our findings suggest that while additional information can have a detrimental effect, they also highlight the challenge of identifying such harmful information. The difficulty in identifying damaging passages primarily arises from the opaque nature of LLMs, which act as black boxes. To the best of my knowledge, no research completely elaborates how language models amalgamate long contextual knowledge with their parametric knowledge. Right now, the community lacks the tools to understand the exact reasons behind these detrimental effects. While we cannot answer the exact reason for why this happens at this moment in time, we hope that our analyses will help show the challenges available and help rally the community to study this important problem. We do answer the question in our title, and importantly, we show that if the problem can be resolved, answer accuracy can be greatly improved for many open-domain knowledge-intensive models while using fewer contextual passages.
>
> **R3) The 10% performance gain is achieved in an ideal scenario. Identifying the definite negative and semi-negatives is not a trivial problem.**
>
> We divided R3 into two sub-questions.
>
> **R3-1) 10% performance gain is achieved in an ideal scenario**
>
> Our findings present the challenges of evaluating this very common architecture. Even with a superset of the evidence, models create mistakes. This is a negative feature that must be studied in future work when building more resilient open-domain NLP systems. It is critical to understand the limitations of the model. From this, we can understand the behaviors and limitations and make improvements to the models that use retrieved evidence. Our method isn't a SOTA model for benchmarking; our goal is to demonstrate this limitation and identify a problem that will motivate future research. We focus on identifying passages that detrimentally affect the final inferences made by the Reader model.
>
> **R3-2) Identifying the definite negative and semi-negatives is not a trivial problem**
> Yes. Differentiating between these definite negative passages and semi-negatives is a challenge and definitely not trivial. One of our research goals is to determine the feasibility of capturing the pattern or signal within Reader that can be learned by a neural network when transition occurs. So, we put more emphasis on DP and DN; however, this does not mean that DN and SN are trivial. We conducted additional experiments to evaluate other methods for identifying the other passage types as well. Details and experiments are reported in Q3-2)
>
> **Q1) Figure 4: DF -> DP in the legend**
>
> We corrected this issue. Thank you.
>
> **Q2) Line 569: This finding supports ? that**
>
> We intended to refer to this paper; [Tomayto, Tomahto. Beyond Token-level Answer Equivalence for Question Answering Evaluation](https://aclanthology.org/2022.emnlp-main.20) (Bulian et al., EMNLP 2022) regarding answer equivalence. Sorry that this link was broken.
>
> **Q3) Why binary classification is considered instead of multi-class classification on the 5 types? Why not apply the classification results to the passage selection to see the end-to-end result?**
>
> We Address Q3 in two different responses.
>
> **Q3-1) Why binary classification is considered instead of multi-class classification on the 5 types?**
>
> We focus on Definite positives (DP) and Definite negative (DN) as these are transitions for the model where it changes the answer. We believe these to be the most significant for classification. If DP and DN can be distinguished through binary classification, this could serve as a foundational step toward implementing the suggested multi-class classification approach. However, we also agree with your opinion that multi-class classification can be applied. We lay out multi-classification on the 5 types with its end-to-end result in Q3-2).
>
> **Q3-2) Why not apply the classification results to the passage selection to see the end-to-end result?**
>
> In the paper, we conclude that DPs and DNs are not easy to distinguish based on their binary classification and attention analysis. Also, as described in Table 5, all the models tend to classify DN as DP. Therefore, directly applying this classification cannot properly perform a selection method since DNs are considered to be DPs. However, based on the feedback, we experiment with two classification methods: binary classification on the whole EM pattern and multi-classification on 5 types (IZ, DP, DN, SP, SN). Also, for each classification method, we evaluate intermediate results (accuracy, f1, recall, precision) and end-to-end performances.
>
> **Binary/Multi Classification**
>
> For all classification methods, the input format is the same as that of FiD’s, “question : {question} title: {title} context : {passage}”. The output is EM pattern, 0 or 1 for binary classification, and context type (IZ, DP, DN, SP, SN) for multi-classification. In contrast to the previous binary classification, the new binary classification is trained on the entire context.
>
> We train three models: RoBERTa(Large), FiD Encoder(Large), and T5 Encoder(Large). RoBERTa(Large) and T5 Encoder(Large) are initialized from the huggingface checkpoints. We employ FiD Encoder(Large) available on FiD's Git repository, pre-trained on NQ datasets. We use the same datasets as for DP/DN binary classification, resulting in 700,500 and 175,200 instances, respectively. For end-to-end evaluation, we use the NQ test set. All training uses a batch size of 64, a learning rate of 5e-5, 11,000 steps (about 1 epoch), and other huggingface default parameters.
>
> The table below describes the intermediate results of binary classification models on the development set.
> | Model | Accuracy | Recall | Precision | F-1   |   |   |   |   |   |
> |------------|----------|--------|-----------|-------|---|---|---|---|---|
> | FiD        | 0.564    | 0.791  | 0.547     | 0.646 |   |   |   |   |   |
> | T5         | 0.578    | 0.638  | 0.573     | 0.604 |   |   |   |   |   |
> | RoBERTa    | 0.503    | 1      | 0.503     | 0.669 |   |   |   |   |   |
>
> The high recall and low precision indicate models often predict 1s even when true labels are 0s. RoBERTa consistently predicts 1s aligns with DP/DN classification. These results show binary classification inadequacy in identifying EM patterns, and performance demonstrates significant variation depending on the model.
>
> |                |           | DN  | SN    | SP    | IZ    | DP   |   |   |   |
> |----------------|-----------|-----|-------|-------|-------|------|---|---|---|
> | FiD            | precision | 0   | 0.09  | 0.54  | 0.5   | 0    |   |   |   |
> |                | recall    | 0   | 0.04  | 0.72  | 0.37  | 0    |   |   |   |
> |                |           | DN  | SN    | SP    | IZ    | DP   |   |   |   |
> | T5            | precision | 0   | 0.1   | 0.55  | 0.5   | 0    |   |   |   |
> |                | recall    | 0   | 0.05  | 0.72  | 0.37  | 0    |   |   |   |
> |                |           | DN  | SN    | SP    | IZ    | DP   |   |   |   |
> | RoBERTa        | precision | 0   | 0     | 0.5   | 0     | 0    |   |   |   |
> |                | recall    | 0   | 0     | 1     | 0     | 0    |   |   |   |
> | Total instance |           | 396 | 15438 | 86936 | 71122 | 1308 |   |   |   |
> |                |           |     |       |       |       |      |   |   |   |
>
> The table shows that models struggle with identifying DN & DP, favoring predictions of SP, which constitutes 50% of instances. Notably, all three models can't differentiate DNs and DPs at all. Among 175,200 instances, T-5 assigns 114,606 (65.4%) to SP, FiD labels 115,522 (65.9%) as SP, and RoBERTa classifies all as SP. This highlights multi-classification's challenge in capturing passage types and inability to identify DN and DP.
>
> **End-to-End Result**
>
> For the end-to-end evaluation, we construct two test sets from the NQ test dataset for each classification method. Models use these test sets to generate either EM patterns or passage types. For EM patterns, the selection method can be directly applied. For passage types, they are first converted to EM patterns before applying the selection method.
>
> The first table presents the Exact Match score for the end-to-end result on the NQ test set. For comparison, the second table displays EM@100 and Probe3 on the same NQ test set.
>
> |        |         | Probe1 | Probe2 | Probe3 | Probe4 | Probe5 | Probe6 |   |   |
> |--------|---------|--------|--------|--------|--------|--------|--------|---|---|
> | Binary | FiD     | 46.18  | 35.37  | 52.69  | 42.27  | 53.66  | 43.3   |   |   |
> |        | T5      | 46.12  | 36.98  | 53.57  | 44.68  | 54.27  | 45.29  |   |   |
> |        | RoBERTa | 54.43  | 35.84  | 54.43  | 35.84  | 54.43  | 35.84  |   |   |
> | Multi  | FiD     | 48.59  | 37.51  | 53.77  | 42.74  | 54.71  | 43.66  |   |   |
> |        | T5      | 48.73  | 37.45  | 53.46  | 42.3   | 54.24  | 43.19  |   |   |
> |        | RoBERTa | 54.43  | 35.84  | 54.43  | 35.84  | 54.43  | 35.84  |   |   |
>
> | EM@100 | Probe 3 |   |   |   |   |   |   |   |   |
> |--------|---------|---|---|---|---|---|---|---|---|
> | 54.43  | 65.12   |   |   |   |   |   |   |   |   |
>
> Among all classification models, both binary RoBERTa and multi RoBERTa achieve the highest EM scores across Probe1, Probe3, and Probe5. This is due to all RoBERTa inferences being 1s or SPs for each classification, resulting in all contexts being used as inputs for the reader in Probe1, 3, and 5. Consequently, these scores should match EM@100, which uses all contexts for inference.
>
> All Probe results exhibit either lower or equal performance compared to EM@100, notably falling significantly below Probe 3's EM score. This underscores the classification models' struggle in identifying EM patterns and passage types, reflecting the challenge in distinguishing DN and DP within the provided retrieved context, as observed in the end-to-end evaluation.
>
> We will integrate these findings into the Ablation studies following the results and analysis

---

### Official Review · Reviewer_5NS3 · 2023-08-04

**Soundness:** 3

**Excitement:**

3: Ambivalent: It has merits (e.g., it reports state-of-the-art results, the idea is nice), but there are key weaknesses (e.g., it describes incremental work), and it can significantly benefit from another round of revision. However, I won't object to accepting it if my co-reviewers champion it.

**Paper Topic And Main Contributions:**

This paper addresses the problem of damaging retrieval for open domain question answering. The authors propose a method for identifying the passages that are detrimental for the QA task by evaluating how different subsets of the retrieved passages change the answer. The experiments use an FiD model and the method is evaluated on the NaturalQuestions and  the TriviaQA datasets.

The authors rely on the Exact Match pattern where  increasingly longer sequences are given as a context to a generative model. The interesting passages produce changes in the EM score of the generated answers from 0 to 1 or from 1 to 0. Based on the EM pattern the passages can be classified  in 5 categories: IZ (Initial Zero), DP (Definite positive), DN(definite negatives), SP(semi-positive) and SN(semi negative).  Once the type of the passages are identified, there are 6 probing strategies that can be used to construct the context from the retrieved passages.
The experiments show that using the DP and SP passage types in the context improved the EM of the generation.

**Questions For The Authors:**

Please see Reasons to Reject

**Reasons To Accept:**

Finding the informative context for the open QA task is an important problem.
The authors show that simple approaches for identifying the evidence in the retrieved passages can be effective.

**Reasons To Reject:**

1. The authors propose a simple approach for evidence selection from the retrieved passages and the novel contributions are rather limited. While some analysis in the paper is interesting, in my opinion the main points could de describe in a short paper.
2. The approach has serious limitations, making it hard to apply to real problems. Relying on the order of retrieved passages raises questions regarding the upper bound performance as well as concerns regarding the generalization of the approach to other domains. The use of the NQ and TQ datasets is also limiting due to the fact that the answers are concise and short. It unclear how the context selection applies for long form QA.
3. The approach relies on using the gold answer for passage selection. This is not possible in real tasks. The use of the binary classifier to identify DP and SP passage types is not effective since it tends to classify all passages as useful (DP).
4. The experimental section could be made more clear. It now reads as a sequence of findings and it is not immediately clear why some experiments are important. I had difficulty relating the passage attention as well as the binary classifier experiments to the main points of the paper.  It would also be useful to specify more clearly what the lower and upper bounds are in section 6.1 .

**Reproducibility:**

3: Could reproduce the results with some difficulty. The settings of parameters are underspecified or subjectively determined; the training/evaluation data are not widely available.

**Reviewer Confidence:**

4: Quite sure. I tried to check the important points carefully. It's unlikely, though conceivable, that I missed something that should affect my ratings.

---

> ### Author Rebuttal · Authors · 2023-08-29
>
> Dear Reviewer, 5NS3
>
> Thank you for the time to review our paper thoroughly. We deeply appreciate how you found our approach to identifying relevant evidence significant, which currently does not fully leverage the retrieved passages.
>
> Below, we have tried our best to address your questions and concerns. In our rebuttal, we have addressed each "Reason to Reject" and "Question for the Authors" as denoted by R1, R2, Q1, Q2, and so forth. If any subsections or questions are present within these, they will be referenced as R1-1, R1-2, and so on. Please take a look at our response and let us know if further clarification is needed.
>
> **R1) The authors propose a simple approach for evidence selection from the retrieved passages and the novel contributions are rather limited. While some analysis in the paper is interesting, in my opinion the main points could de describe in a short paper.
> .**
>
> The contribution of the paper isn't just the methodology. We introduce a novel concept, the notion of damaging passages in the context of the question-answering task that has yet to be thoroughly explored. Thus needs space for analysis. Here is the overview of our work:
> - Introducing the knowledge-intensive NLP, Retrieval, Damaging Retrieval
>     - We introduce the domains of knowledge-intensive NLP, retrieval methodologies, and the concept of damaging retrieval.
> - Simulating damaging passages in question answering
>     - Our study simulates damaging passages within the context of question answering.
> - Demonstration of Order-Invariant FiD
>     - We underscore the order-invariant property of FiD
> - Incremental Inference using FiD
>     - Building upon the aforementioned insights, we conduct incremental inference utilizing FiD as the reader, unveiling the presence of detrimental effects.
> - Propose Selection Inference
>      - We introduce the concept of selection inference and confirm the existence of subsets within retrieved passages that exhibit higher Exact Match (EM) scores than conventional EM computed over the entire retrieved list.
> - Analysis of DP and DN
>     - Our study employs two distinct methodologies to analyze transitions triggered by DP and DN passages.
>     - Attention Score Analysis
>         - We delve into attention score analysis, a technique previously employed to enhance reader performance and elucidate that it fails to reveal transition signals effectively.
>     - Evaluation of Binary Classification for Transition Pattern
>         - We assess the efficacy of binary classification in capturing transition patterns.
>
> We can be a bit more terse with the explanation, making more space for justifying our method, adding further analyses, and improving the organization of the paper.
>
> **R2) The approach has serious limitations, making it hard to apply to real problems. Relying on the order of retrieved passages raises questions regarding the upper bound performance as well as concerns regarding the generalization of the approach to other domains. The use of the NQ and TQ datasets is also limiting due to the fact that the answers are concise and short. It unclear how the context selection applies for long form QA.**
>
> We address R2 in two different responses R2-1), and R2-2).
>
> **R2-1) Relying on the order of retrieved passages raises questions regarding the upper bound performance as well as concerns regarding the generalization of the approach to other domains.**
>
> Yes. Our findings rely on the order of retrieved passages. There are two main reasons for utilizing the order from the retriever: 1) Top passages extracted by these retrievers are highly likely to contain information relevant to the answer by its nature, and 2) True optimal upper bound for retrieved passages is intractable. Here, we illustrate each reason in detail.
>
>
> - Recall-optimized retrievers
>
>     Retrievers are optimized for recall. This implies that passages ranked higher contain more pertinent information to the answer than those ranked lower. This aligns with the results depicted in Figures 6,7, and 8. All retrievers present a steady decrease in the increment rate of AcEM and EM with respect to the number of passages. This saturation in AcEM and EM implies that lower-ranked passages offer little information about the answer.
>
> - Intractability of the true upper bound
>
>     Determining the true upper bound is not feasible in practice. Consider a scenario with 100 retrieved passages. Let's denote k as the number of selected passages used for inference. For a single query and corresponding 100 passages, it would require 100! times k inferences for k ranging from 1 to 100  to establish the optimal upper bound. Given that our paper demonstrates the order invariant property of FiD, this could be further reduced to 100Ck for k ranging from 1 to 100 where 100Ck pertains to the selection of k passages. However, despite these considerations, this still remains practically intractable.
>
>     On the other hand, calculating AcEM based on the retrieved order is achievable, requiring 100 inferences for a single question and retrieved passages. For our research, the computation of AcEM sets requires approximately 100 times more effort than that for the vanilla model, accounting for the subsets of evidence.
>
> Based on our findings, we demonstrate that a higher upper bound can be established given a retrieved passage through AcEM, which illustrates the limitation of the previous upper bound formed by  EM match score.
>
> **R2-2) The use of the NQ and TQ datasets is also limiting due to the fact that the answers are concise and short. It unclear how the context selection applies for long form QA.**
>
> We disagree with the assertion that the use of NQ and TQ datasets in our findings undermines the potential future research in long form QA. Instead, our work establishes the foundation for investigating the influence of damaging passages in long-form QA, for the following reasons:
>
> - Prior works in damaging passages revolve around simple tasks, such as fact verification and query operations, where responses are specific and confined. Despite the simplicity of these tasks, the presence of relevant passages has been shown to have negative consequences.
>
> - Employment of the NQ and TQ datasets aims to identify the presence of detrimental passages in question answering tasks. Through the application of EM patterns, we substantiate the existence of damaging passages within question answering. Building upon this observation, we apply selection inference, showing the model's performance can be enhanced.
>
> - Direct application of the selection method to long-form QA might be challenging because answers are in the form of sentences, and there is no deterministic metric such as EM. However, this does not mean that the damaging passages cannot be captured. If we confine the number of contexts and their input sizes, we may perform human evaluations to verify the existence of damaging effects in long-form QA. Further, we can conduct incremental inference and qualitative analysis, as we conducted in the paper, to confirm the existence of damaging passages in long-form QA.
>
> **R3) The approach relies on using the gold answer for passage selection. This is not possible in real tasks. The use of the binary classifier to identify DP and SP passage types is not effective since it tends to classify all passages as useful (DP).**
>
> Yes. The approach relies on employing the gold answer for passage selection. Nonetheless, it is worth noting that our method isn't intended as a SOTA model for benchmarking purposes. Instead, we aim to underscore this limitation and identify a problem that will inspire future research. Detecting damaging passages is a challenge that demands future efforts when building more resilient open-domain NLP systems. From this, we can understand the behaviors and limitations and improve the models that use retrieved evidence.
>
> Our contributions are sound and clear.
>
> - Contrary to conventional assumption, we verify that extra relevant passages can degrade the model's performance in question answering in a real-world setting.
> - We identified the existence of subsets from the retrieved passages that can generate higher EM scores without any architectural modifications.
> - We demonstrated the complexity of distinguishing between DP and DN, which requires further research.
>
> To conclude, we denote to a recently published paper, [Lost in the Middle: How Language Models Use Long Contexts](https://arxiv.org/abs/2307.03172)  (Liu, Nelson F., et al.,2023), also investigating the identification of relevant passage within long inputs. This paper contains a section, ‘Is More Context Always Better?’. (This paper was published after the EMNLP submission deadline). In this section, they said NO, referring to the saturation point in the Reader's accuracy with respect to the number of passages. They suggest that reordering of retrieved documents and truncation of ranked lists are promising directions for future work. However, our research can provide different insights, filtering out damaging passages for higher inference results as well as run-time efficiency.
>
> **R4) The experimental section could be made more clear. It now reads as a sequence of findings and it is not immediately clear why some experiments are important. I had difficulty relating the passage attention as well as the binary classifier experiments to the main points of the paper. It would also be useful to specify more clearly what the lower and upper bounds are in section 6..**
>
> We address R4 in two different responses.
>
> **R4-1) The experimental section could be made more clear. It now reads as a sequence of findings and it is not immediately clear why some experiments are important. I had difficulty relating the passage attention as well as the binary classifier experiments to the main points of the paper.**
>
> We appreciate the feedback. To enhance clarity, we will explain the motivations and context of attention analysis and binary inference in section 4 and offer more detailed settings in section 5. Furthermore, we will clearly separate the experimental findings and further analyses. We will implement as follows:
>
> - Explain previous works and how they relate to the main points of the paper for attention analysis and binary classification
> - For the experiment section, we will articulate the experimental settings in more details
>
> **R4-2) It would also be useful to specify more clearly what the lower and upper bounds are in section 6**
>
> We will replace the lower bounds and upper bounds with EM@100, AcEM@100. We will also articulate that EM@100 refers to the highest Exact Match Score of the Reader utilizing top-100 passages, and AcEM@100 denotes incremental inference results on the same top-100 passages in Table 2 as well.

---

### Official Review · Reviewer_RWj7 · 2023-08-07

**Soundness:** 4

**Excitement:**

3: Ambivalent: It has merits (e.g., it reports state-of-the-art results, the idea is nice), but there are key weaknesses (e.g., it describes incremental work), and it can significantly benefit from another round of revision. However, I won't object to accepting it if my co-reviewers champion it.

**Paper Topic And Main Contributions:**

The paper focuses on the open-domain QA task. In particular, the extent to which the performance of architectures based on the Fusion-in-Decoder (FiD) is impacted by the relevance and the number of the input passages, which are provided by a retriever model. The authors provide an analysis about the conditions under which the resulting answer from an FiD-based architecture can be improved should any harmful passages (i.e. passages whose inclusion would be harmful for the Exact Match score) are omitted from the reader model's input. Furthermore, they explore different approaches for identifying harmful passages dynamically (i.e. by using either a binary classifier or heuristics based on the reader model's attention scores), without requiring hypothesising the existence of gold-answer data.

**Questions For The Authors:**

Question A: Which retriever is used for the experiments presented in Section 6.2 (and Table 3)?
Question B: What would be compression rate (i.e. the average percentage drop with respect to the number of used passages) that is achieved by using the five passages of probe 3 in comparison to the number of passages that are used by the upper-bound solution of Table 2?

**Reasons To Accept:**

- Interesting methodology for identifying different passage types along with the relevant probe patterns (i.e. combinations of different passage types that will be used as input to the reader model).
- The domain of the work is interesting. The paper lays the groundwork for understanding better the detrimental effect that some passages can have on the performance of FiD-based architectures.
- Different insights and solutions are explored for seeking to identify automatically passages whose inclusion in the reader model's input would be harmful for the final answer.

**Reasons To Reject:**

- I think the paper could benefit from another round of proof-reading. There are several sections that are difficult to follow.
- My understanding is that the solution for identifying the passage type hypothesises the existence of gold-answer data. The paper goes one-step further and highlights the challenges associated with seeking to identify harmful passages, when gold-answer data is not available. However, no concrete solutions to this problem are provided.

**Reproducibility:**

3: Could reproduce the results with some difficulty. The settings of parameters are underspecified or subjectively determined; the training/evaluation data are not widely available.

**Reviewer Confidence:**

3: Pretty sure, but there's a chance I missed something. Although I have a good feel for this area in general, I did not carefully check the paper's details, e.g., the math, experimental design, or novelty.

**Typos Grammar Style And Presentation Improvements:**

- "put" in line 59 should be replaced with "but".
- The order of the figures is not consistent with the train-of-thought presented in the main body of the manuscript. For instance, Figure 1 (located at page 1) is first referred to, in the main body, at the end of page 4, while Figure 2 (located at page 4) at the middle of page 3.
- There are parts of the paper that could benefit from rephrasing. For instance: lines 76--85 and lines 86--87 from introduction and lines 234--240 from Section 3.1.
- Something (maybe "passages") is missing from line 295 of Section 3.2, after "top-K".
- The legend of Figure 4 refers to a non-existent type of passage (i.e. "DF"). Should it be replaced by "DP"?
- Some captions, such as those of Figure 3 and 4, are also lacking proper punctuation.

---

> ### Author Rebuttal · Authors · 2023-08-29
>
> Dear Reviewer RWj7,
>
> We sincerely appreciate your thoughtful review. We are glad that you find our attempt to filter out damaging passages laying the groundwork for better understanding the detrimental effect of some passages on FiD-based architecture.
>
>
> Below, we have tried our best to address your questions and concerns. We have addressed each "Reason to Reject" and "Question for the Authors" as denoted by R1, R2, Q1, Q2, and so forth. If any subsections or questions are present within these, they will be referenced as R1-1, R1-2, and so on. Regarding "Typos Grammar Style And Presentation Improvements," we will use T1 and T2 for reference. Please take a look at our response and let us know if further clarification is needed.
>
> **R1) I think the paper could benefit from another round of proof-reading. There are several sections that are difficult to follow.**
>
> We value your suggestion. We are currently undergoing a proof-reading to enhance the organization based on feedback from reviewers. Our goal is to provide readers with a clearer understanding of our work.  Additionally, as suggested in the presentation improvement section, we have rephrased specific portions of the paper. These revisions can be found in T3).
>
> **R2) My understanding is that t`he solution for identifying the passage type hypothesises the existence of gold-answer data. The paper goes one-step further and highlights the challenges associated with seeking to identify harmful passages, when gold-answer data is not available. However, no concrete solutions to this problem are provided.**
>
> We want to point out that the absence of a specific solution to the problem does not diminish the value of our work. Our objective is to emphasize this limitation and identify a problem that will drive future research. Detecting damaging passages is a challenge that demands future efforts when building more resilient open-domain NLP systems. From this, we can understand the behaviors and limitations and make improvements to the models that use retrieved evidence.
>
> Our contributions are sound and clear.
> - Contrary to conventional assumption, we introduced extra relevant passages that can degrade the performance of the model in question answering tasks.
> - We identified the existence of subsets from the retrieved passages with higher EM scores without any architectural modifications.
> - Also demonstrated the complexity of distinguishing between DP and DN, which requires further research.
>
> To conclude, we denote to a recently published paper, [Lost in the Middle: How Language Models Use Long Contexts](https://arxiv.org/abs/2307.03172)  (Liu, Nelson F., et al.,2023), also investigating the identification of relevant passage within long inputs. This paper contains a section, ‘Is More Context Always Better?’. (This paper was published after the EMNLP submission deadline). In this section, they said NO for the question, referring to the saturation point in the Reader's accuracy with respect to the number of passages. They suggest that the reordering of retrieved documents and truncation of ranked lists are promising directions for future work. However, our research can provide different insights, filtering out damaging passages for higher inference results as well as run-time efficiency.
>
> **Q-A) Question A: Which retriever is used for the experiments presented in Section 6.2 (and Table 3)?**
>
> We use DPR(Dense Passage Retriever) provided by the authors for the experiments presented in Section 6.2 (and Table 3). We will add this information in Section 6.2 as well as Table 3.
>
> **Q-B) What would be compression rate (i.e. the average percentage drop with respect to the number of used passages) that is achieved by using the five passages of probe 3 in comparison to the number of passages that are used by the upper-bound solution of Table 2?**
>
> We introduce three additional columns (Probe3@5, EM@5, Compression rate) to Table 2. To enhance clarity, we replace “Lower” and “Upper” with “EM@100” and “AcEM100”, respectively. We explain each column component as follows:
> - EM@100 refers to the highest Exact Match Score of the Reader utilizing top-100 passages
> - AcEM@100 denotes incremental inference results on the same top-100 passages.
> - Probe 3 presents the inference outcome of the selection method outlined in Figure 5.
> - Probe 3@5 denotes the inference outcome of utilizing top-5 passages in Probe 3.
> - Compression rate is derived by subtracting Probe 3@5 from AcEM100 and dividing by the number of passages used (5).
> - EM@5 represents the inference outcome using top-5 passages from the retrieved list
>
> | Dataset | Retreiver  | EM@100 | AcEM100 | Probe 3 | Probe 3@5 | EM@5  | Compression rate |   |   |
> |---------|------------|--------|---------|---------|-----------|-------|------------------|---|---|
> | NQ      | DPR        | 52.5   | 62.3    | 61.8    | 57.55     | 45.8  | 0.95             |   |   |
> |         | SEAL       | 50     | 59.4    | 52.9    | 46.21     | 38.47 | 2.63             |   |   |
> |         | Contriever | 50.7   | 60.8    | 53      | 46.09     | 40.02 | 2.94            |   |   |
> | TQA     | DPR        | 72.3   | 77.7    | 77.6    | 75.77     | 66.45 | 0.38            |   |   |
> |         | SEAL       | 67.1   | 72.3    | 72.5    | 66.48     | 55.26 | 1.16            |   |   |
> |         | Contriever | 69.7   | 75.5    | 75.5    | 66.56     | 60.14 | 2.78             |   |   |
>
> When evaluating the NQ dataset, we utilize the FiD-large reader that was trained on the DPR-retrieved NQ training dataset. Similarly, for the TQA dataset, we employ the FiD-large reader trained on the DPR-retrieved TQA training dataset.
>
> **T1), T4), T5), T6) Typos**
>
> - T1) "put" in line 59 should be replaced with "but".
>     -  Right.
> - T4)  Something (maybe "passages") is missing from line 295 of Section 3.2, after "top-K".
>     - Right. "Passages" is missing.
> - T5) The legend of Figure 4 refers to a non-existent type of passage (i.e. "DF"). Should it be replaced by "DP"?
>     - Yes, it should be replaced by "DP".
> - T6) Some captions, such as those of Figure 3 and 4, are also lacking proper punctuation.
>     - We will keep proper punctuation.
>
>
> We really appreciate your pointing out typos.
>
> **T2) The order of the figures is not consistent with the train-of-thought presented in the main body of the manuscript. For instance, Figure 1 (located at page 1) is first referred to, in the main body, at the end of page 4, while Figure 2 (located at page 4) at the middle of page 3.**
>
> T2) Figure 1 is located on page 1 to give readers a general overview of our paper. We will reference Figure 1 in the introduction section on page 1 to enhance its clarity. As for the remaining figures and tables, we will ensure they are appropriately aligned with their respective reference points. We appreciate your valuable suggestions.
>
> **T3) There are parts of the paper that could benefit from rephrasing. For instance: lines 76--85 and lines 86--87 from introduction and lines 234--240 from Section 3.1.**
>
> **T3-1) We have rephrased lines 76 ~ 85 to emphasize our contributions and findings**
>
> [Preivous] Identifying context passages that are detrimental to the performance of the model, and removing them illustrates performance bounds for the models under ideal conditions showing how different subsets of the retrieved evidence result in up to a 10% improvement in exact match score: this exceeds the performance gains from contemporaneous works proposing extensions to FiD such as FiD-light and fusion in encoder
>
> [Rephrased] In summary, our findings challenge the conventional assumption that more passages lead to higher performance. In fact, introducing additional relevant context can actually worsen the performance of the model in question answering. By excluding passages that have a detrimental effect, we observe up to 10% improvement in the exact match score under ideal conditions. This enhancement is achieved by identifying subsets within the retrieved passages that yield higher EM scores without requiring any architectural modifications. Notably, these gains surpass the performance improvements seen in recent works that propose FiD extensions like FiD-light and fusion in encoder.
>
>
> **T3-2) We have rephrased lines 86~87 to emphasize our findings**
>
> [Preivous] Even though models have access to a sufficient set of information to generate the answer, the a superset of this information is detrimental.
>
> [Rephrased] Our contributions reveal how to leverage the retrieved passages. This involves not only identifying pertinent passages but also excluding harmful ones, highlighting the importance of filtering out damaging passages.
>
> **T3-3) We have rephrased lines 234--240 from Section 3.1 to emphasize our contributions in a more concise manner down below**
>
> [Preivous] Given this perspective, our primary focus and contribution for this paper is on removing passages that may harm the overall accuracy of the system, but rather than simulating these from a different retrieval model, our goal is to identify these from the same retrieval model used for the full pipeline rather than simulating them.
>
> [Reprhased] One of the contributions in this paper is the removal of damaging passages, which can undermine the accuracy of the system in open-domain question answering tasks where damaging effects have not been thoroughly investigated. Ultimately, our objective is to validate the presence of detrimental effects within the complete open-domain pipeline rather than relying solely on simulations.
>
> We will rephrase parts of the paper if needed to give a clear understanding and explanation to readers.

---

### Meta-Review · Area_Chair_ptrY · 2023-09-19

**Recommendation:** 3

**Metareview:**

The reviewers are in disagreement about the soundness and exciting qualities of this work, giving scores between ambivalent and strong.

Questions about soundness centered on the goals of the study. Reviewers critiqued the practical applicability of the method (e.g., when gold data is not available), but the authors clarified that their goal was not to build a universal tool for model improvement but a higher level goal of problem illumination. I believe the authors adequately responded to these concerns, which were the strongest of the soundness concerns.

The simplicity of their method is a pro, and and the various insights and tests seeking harmful passages are useful in themselves and can inspire future research. These insights contribute to some reviewers finding the work strongly exciting.

The reviewers suggested clarifying the organization and writing in some sections of the paper, and these change would greatly improve a final version of the paper. I agree with this request and believe the authors can make the paper clearer (as one example: by labeling axes on plots and providing more intuitive explanations in figure captions).

---

### Decision · Program_Chairs · 2023-10-07

**Decision:**

Accept-Findings

**Comment:**

The reviewers are in disagreement about the soundness and exciting qualities of this work, giving scores between ambivalent and strong.

Questions about soundness centered on the goals of the study. Reviewers critiqued the practical applicability of the method (e.g., when gold data is not available), but the authors clarified that their goal was not to build a universal tool for model improvement but a higher level goal of problem illumination. I believe the authors adequately responded to these concerns, which were the strongest of the soundness concerns.

The simplicity of their method is a pro, and and the various insights and tests seeking harmful passages are useful in themselves and can inspire future research. These insights contribute to some reviewers finding the work strongly exciting.

The reviewers suggested clarifying the organization and writing in some sections of the paper, and these change would greatly improve a final version of the paper. I agree with this request and believe the authors can make the paper clearer (as one example: by labeling axes on plots and providing more intuitive explanations in figure captions).